# LEARNING COLLISION-FREE LATENT SPACE FOR BAYESIAN OPTIMIZATION

## ABSTRACT

Learning and optimizing a blackbox function is a common task in Bayesian optimization and experimental design. In real-world scenarios (e.g., tuning hyper-parameters for deep learning models, synthesizing a protein sequence, etc.), these functions tend to be expensive to evaluate and often rely on high-dimensional inputs. While classical Bayesian optimization algorithms struggle in handling the scale and complexity of modern experimental design tasks, recent works attempt to get around this issue by applying neural networks ahead of the Gaussian process to learn a (low-dimensional) latent representation. We show that such learned representation often leads to *collision* in the latent space: two points with significantly different observations *collide* in the learned latent space. Collisions could be regarded as additional noise introduced by the traditional neural network, leading to degraded optimization performance. To address this issue, we propose *Collision-Free Latent Space Optimization* (CoFLO), which employs a novel regularizer to reduce the collision in the learned latent space and encourage the mapping from the latent space to objective value to be Lipschitz continuous. CoFLO takes in pairs of data points and penalizes those too close in the latent space compared to their target space distance. We provide a rigorous theoretical justification for the regularizer by inspecting the regret of the proposed algorithm. Our empirical results further demonstrate the effectiveness of CoFLO on several synthetic and real-world Bayesian optimization tasks, including a case study for computational cosmic experimental design.

## 1 INTRODUCTION

Bayesian optimization is a classical sequential optimization method and is widely used in various fields, including recommender systems, scientific experimental design, hyper-parameter optimization, etc. Many of theses applications involve evaluating an expensive blackbox function; therefore the number of queries should be minimized. A common way to model the unknown function is via Gaussian processes (GPs) Rasmussen and Williams (2006). GPs have been extensively studied under the bandit setting, and has proven to be an effective approach for addressing a broad class of black-box function optimization problems. One of the key computational challenges for learning with GPs concerns with optimizing specific kernels used to model the covariance structures of GPs. As such optimization task depends on the dimension of feature space, for high dimensional input, it is often prohibitively expensive to train a Gaussian process model. Meanwhile, Gaussian processes are not intrinsically designed to deal with structured input that has a strong correlations among different dimensions, e.g., the graphs and time sequences. Therefore, dimensionality reduction algorithms are needed to speed up the learning process.

Recently, it has become popular to investigate GPs in the context of latent space models. As an example, deep kernel learning (Wilson et al., 2016) simultaneously learns a (low-dimensional) data representation and a scalable kernel via an end-to-end trainable deep neural network. In general, the neural network is trained to learn a simpler latent representation with reduced dimension and has the structure information already embedded for the Gaussian process. Such a combination of neural network and Gaussian process could improve the scalability and extensibility of classical Bayesian optimization, but it also poses new challenges for the optimization task (Tripp et al., 2020). As we later demonstrate, one critical challenge brought by introducing the neural network is that the latent representation is prone to *collisions*: two points with significant different observations can get

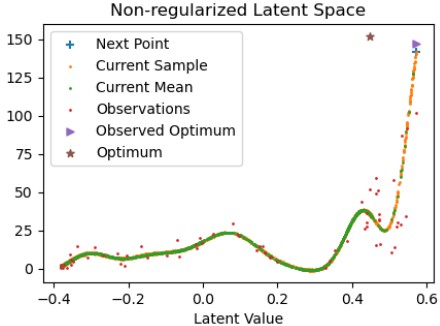

Figure 1: Illustration of the *collision effect* in latent space-based Bayesian optimization tasks. Since the data points around the optimum severely collided, BO is misguided to the sub-optimum.

too close in the latent space. The collision effect is especially evident when information is lost by dimension reduction, and/or when the training data is limited in size in Bayesian optimization.

As illustrated in Figure 1, when passed through the neural network, data points with drastically different observations are mapped to close positions in the latent space. Such collisions could be regarded as additional noise introduced by the neural network. Although Bayesian optimization is known to be robust to mild noisy observations, the collision in latent space could be harmful to the optimization performance, as it is non-trivial to explicitly model the collision into the acquisition function. In addition, the additional noise induced by the collision effect will further loosen the regret bound for classical Bayesian optimization algorithms (Srinivas et al., 2010).

**Overview of main results**  To mitigate the collision effect, we propose a novel regularization scheme which can be applied as a simple plugin amendment for the latent space-based Bayesian optimization models. The proposed algorithm, namely Collision-Free Latent Space Optimization (CoFLO), leverages a regularized regression loss function, to periodically optimize the latent space for Bayesian optimization.

Concretely, our regularizer is encoded by a novel *pairwise collision penalty* function defined jointly on the latent space and the output domain. In order to mitigate the risk of collision in the latent space (and consequently boost the optimization performance), one can apply the regularizer uniformly to the latent space to minimize the collisions. However, in Bayesian global optimization tasks, we seek to prioritize the regions close to the possible optimum, as collisions in these regions are more likely to mislead the optimization algorithm. Based on this insight, we propose a optimization-aware regularization scheme, where we assign a higher weight for the collision penalty on those pairs of points closer to the optimum region in the latent space. This algorithm—which we refer to as dynamically-weighted CoFLO—is designed to dynamically assess the importance of a collision during optimization. Comparing to the uniform collision penalty over the latent space, the dynamic weighting mechanism has demonstrated drastic improvement over the state-of-the-art latent space-based Bayesian optimization models.

We summarize our the key contributions below:

- We propose a novel regularization scheme, as a simple plugin amendment for latent space-based Bayesian optimization models. Our regularizer penalizes collisions in the latent space and effectively reduces the collision effect.

- We propose an optimization-aware dynamic weighting mechanism for adjusting the collision penalty to further improve the effectiveness of regularization for Bayesian optimization.

- We provide theoretical analysis for the performance of Bayesian optimization on regularized latent space.

- We conducted an extensive empirical study on four synthetic and real-world datasets, including a real-world case study for cosmic experimental design, and demonstrate strong empirical performance for our algorithm.

## 2 RELATED WORK

Bayesian optimization has demonstrated promising performance in various cost-sensitive global optimization tasks (Shahriari et al., 2016). However, due to its intrinsic computational limitation in the high-dimensional regime, its applicability has been restricted to relatively simple tasks. In this section, we provide a short survey on recent work in Bayesian learning, which were designed to overcome the high-dimensionality challenge for both Bayesian optimization and regression tasks.

**Deep kernel learning**   Deep kernel learning (DKL) (Wilson et al., 2016) combines the power of the Gaussian process and that of neural network by introducing a deep neural network $g$ to learn a mapping $g : \mathcal{X} \to \mathcal{Z}$ from the input domain $\mathcal{X}$ to a latent space $\mathcal{Z}$, and use the latent representation $z \in \mathcal{Z}$ as the input of the Gaussian process. The neural network $g$ and a spectral mixture base kernel $k$ forms a scalable expressive closed-form covariance kernel, denoted by $k_{DK}(x_i, x_j) \to k(g(x_i), g(x_j))$, for Gaussian processes. Despite of encouraging results in numerous regression tasks, it remains unclear whether DKL is readily applicable to Bayesian optimization. One key difference in Bayesian regression and optimization tasks is the assumption on the accessibility of training data: Bayesian optimization often assumes limited access to labeled data, while DKL for regression relies on abundant access to data in order to train a deep kernel function. Another problem lies in the difference between the objective functionn. While DKL focuses on improving the general regression performance, it does not specifically address the problem caused by collisions, which—as we later demonstrate in section 3.3—could be harmful for sequential decision making tasks.

**Representation learning and latent space optimization**   Aiming at improving the scalability and extensibility of the Gaussian process, various methods are proposed to reduce the dimensionality of the original input. Djolonga et al. (2013) assume that only a subset of input dimensions varies, and the kernel is smooth (i.e. with bounded RKHS norm). Under these assumptions, they underlying subspace via low-rank matrix completion. Huang et al. (2015) use Autoencoder to learn a low-dimensional representation of the inputs to increase GP's scalability in regression tasks. Snoek et al. (2015) further propose to learn a pre-trained encoder neural network before BO. Lu et al. (2018) learn a variational auto-encoder iteratively during sequential optimization to embed the structure of the input. The challenge for combining latent space learning with Bayesian optimization lies in that a pre-trained neural network may not extract adequate information around the more promising region of the input space. Furthermore, the latent space could be outdated without continuous updates with the latest acquired observation. Tripp et al. (2020) propose to periodically retrain the neural network to learn a better latent space, in order to minimize the number of iterations needed for LSO. They claim that by prioritizing the loss of more promising data points in the original input space (i.e. by assigning a higher weight to these data points), the model could focus more on learning high-value regions and allow a substantial extrapolation in the latent space to accelerate the optimization. However, such a framework does not explicitly deal with collisions in the latent space, which we found to be a key factor in the poor performance of modern latent space optimization algorithms.

## 3 PROBLEM STATEMENT

In this section, we introduce necessary notations and formally state the problem. We focus on the problem of sequentially optimizing the function $f : \mathcal{X} \to \mathbb{R}$, where $\mathcal{X} \subseteq \mathbb{R}^d$ is the input domain. In each round $t$, we pick a point $x_t \in \mathcal{X}$, and observe the function value perturbed by an additive noise: $y_t = f(x_t) + \epsilon_t$ with $\epsilon_t \sim \mathcal{N}(0, \sigma^2)$ being i.i.d. Gaussian noise. Our goal is to maximize the sum of rewards $\sum_{t=1}^{T} f(x_t)$ over $T$ iterations, or equivalently, to minimize the *cumulative regret* $R_T := \sum_{t}^{T} r_t$, where $r_t := \max_{x \in \mathcal{X}} f(x) - f(x_t)$ denote the instantaneous regret of $x_t$.

### 3.1 BAYESIAN OPTIMIZATION

Bayesian optimization typically employs Gaussian processes as the statistic tool for modeling the unknown objective function. The major advantage of using GP is that it presents a computationally tractable solution to depict a sophisticated and consistent view across the space of all possible function (Rasmussen and Williams, 2005), which allows closed-form posterior estimation in the function space. BO methods starts with a prior on the black-box function. Upon observing new

labels, BO then iteratively updates the posterior distribution in the function space, and maximizes an acquisition function measuring each point's contribution to finding the optimum, in order to select the next point for evaluation.

Formally, in Bayesian optimization we assume that $f$ follows a $\mathcal{GP}(m(x), k(x, x'))$, where $m(x)$ is the mean function, $k(x, x')$ is the kernel or covariance function. Throughout this paper, we use squared exponential kernel, $k_{SE}(x, x') = \sigma_{SE}^2 \exp\left(-\frac{(x-x')}{2l}\right)$, where the length scale $l$ determines the length of the "wiggles" and the output variance $\sigma_{SE}^2$ determines the average distance of the function away from its mean. At iteration $T$, given the historically selected points $A_T = \{x_1, ..., x_t\}$ and the corresponding noisy evaluations $\mathbf{y}_T = [y_1, ...y_T]$, the posterior over $f$ also takes the form of a GP, with mean $\mu_T(x)$, covariance $k_T(x, x')$, and variance $\sigma_T^2(x)$:

$$\mu_T(x) = k_T(x)^T (K_T + \sigma^2 I)^{-1} \mathbf{y}_T$$
$$k_T(x, x') = k(x, x') - k_T(x)T(K_T + \sigma^2 I)^{-1} k_T(x')$$
$$\sigma_T^2(x) = k_T(x, x)$$

where $k_T(x) = [k(x_1, x), ..., k(x_T, x)]^T$ and $K_T$ is the positive definite kernel matrix $[k(x, x')]_{x, x' \in A_T}$. After obtaining the posterior, one can compute the acquisition function $\alpha : \mathcal{X} \to \mathbb{R}$, which is used to select the next point to be evaluated. Various acquisition functions have been proposed in the literature, including popular choices such as Upper Confidence Bound (UCB) (Srinivas et al., 2010) and Thompson sampling (TS) (THOMPSON, 1933). UCB uses the upper confidence bound $\alpha_{UCB}(x) = \mu_t(x) + \beta^{1/2} \sigma_t(x)$ with $\beta(x)$ being the confidence coefficient, and enjoys rigorous sublinear regret bound. TS usually outperform UCB in practice and has been shown to enjoy a similar regret bound Agrawal and Goyal (2012). It samples a function $\tilde{f}_t$ from the GP posterior $\tilde{f}_t \sim \mathcal{GP}(\mu_t(x), k_t(x, x'))$ and then uses the sample as an acquisition function: $\alpha_{TS}(x) = \tilde{f}_t(x)$.

**Remark.** *Regret is commonly used as performance metric for BO methods. In this work we focus on the simple regret $r_T^* = \max\limits_{x \in \mathcal{X}} f(x) - \max\limits_{t < T} f(x_t)$ and cumulative regret $R(T) = \sum_t^T r_t$.*

### 3.2 LATENT SPACE OPTIMIZATION

Recently, Latent Space Optimization (LSO) has been proposed to solve Bayesian optimization problems on complex input domains (Tripp et al., 2020). LSO first learns a latent space mapping $g : \mathcal{X} \to \mathcal{Z}$ to convert the input space $\mathcal{X}$ to the latent space $\mathcal{Z}$. Then, it constructs an objective mapping $h : \mathcal{Z} \to \mathbb{R}$ such that $f(x) \approx h(g(x)), \forall z \in \mathcal{Z}$. The latent space mapping $g$ and base kernel $k$ could be regarded as a *deep kernel*, denote by $k_{nn}(x, x') = k(g(x), g(x'))$. Thus, the actual input space for BO is the latent space $\mathcal{Z}$ and the objective function is $h$. With acquisition function $\alpha_{nn}(x) := \alpha(g(x))$, it is unnecessary to compute an inverse mapping $g^{-1}$ as discussed in Tripp et al. (2020), as BO could directly select $x_t = \arg\max\limits_{x_t \in \mathcal{X}} \alpha_{nn}(x) \forall t \leq T$ and evaluate $f$. In the meantime, BO can leverage the latent space mapping $g$, usually represented by a neural network, to effectively learn and optimize the target function $h$ on a lower-dimension input space.

### 3.3 THE COLLISION EFFECT OF LSO

When the mapping $g : \mathcal{X} \to \mathcal{Z}$ is represented by a neural network, it may cause undesirable *collisions* between different input points in the latent space $\mathcal{Z}$. Under the noise-free setting, we say there exists a *collision* in $\mathcal{Z}$, if $\exists x_i, x_j \in \mathcal{X}$, such that when $g(x_i) = g(x_j)$, $|f(x_i) - f(x_j)| > 0$. Such collision could be regarded as additional (unknown) noise on the observations introduced by the neural network $g$. For a noisy observation $y = f(x) + \epsilon$, we define a collision as follows: for $\rho > 0, \exists x_i, x_j \in D, |g(x_i) - g(x_j)| < \rho|y_i - y_j|$.

When the distance between a pair of points in the latent space is too close comparing to their difference in the output space, the different output values for the collided points in the latent space could be interpreted as the effect of additional observation noise. In general, collisions could degrade the performance of LSO. Since the collision effect is *a priori* unknown, it is often challenging to deal with collisions in LSO, even if we regard it as additional observation noise and increase the (default)

noise variance in the Gaussian process. Thus, it is necessary to mitigate the collision effect, by directly restraining it in the representation learning phase.

# 4 COLLISION-FREE LATENT SPACE OPTIMIZATION

In this section, we introduce Collision-Free Latent Space Optimization (CoFLO), an algorithmic framework designed to mitigate the collision effect.

## 4.1 OVERVIEW OF THE COFLO ALGORITHM

The major challenge in restraining collisions in the latent space is that, unlike the traditional regression problem, we cannot quantify it on a single point's observation. We can, however, quantify collisions by grouping pairs of data points and inspecting their corresponding observations.

We define the *collision penalty* based on pairs of inputs, and further introduce a pair loss function to characterize the collision effect. Based on this pair loss, we propose a novel regularized latent space optimization algorithm[1], as summarized in Algorithm 1. The proposed algorithm first uses the pair-wise input and concurrently feeds them into the same network and then calculates the pair loss function. We demonstrate this process in Figure 2.

Given a set of labeled data points, we can train the neural network to create an initial latent space representation[2], similar to DKL (Wilson et al., 2016). Once provided with the initial representation, we can then refine the latent space by running CoFLO and periodically update the latent space (i.e. updating the latent representation after collection a batch of data points) to mitigate the collision effect as we collect more labels

---

**Algorithm 1** Collision-Regularized Latent Space Optimization (CoFLO)

---

1: **Input**: Regularization weight $\rho$ (cf. Equation 3), penalty parameter $\lambda$ (cf. Equation 1), retrain interval $\tilde{T}$, importance weight parameter $\gamma$ (cf. Equation 2), neural network $M_0$, base kernel $K_0$, prior mean $\mu_0$, total time steps $T$;
2: **for** $t = 1$ *to* $T$ **do**
3:     $x_t \leftarrow \underset{x \in D}{\arg\max}\, \alpha(M_t(x))$             ▷ *maximize acquisition function*
4:     $y_t \leftarrow$ evaluation on $x_t$             ▷ *update observation*
5:     **if** $t \equiv 0 \pmod{\tilde{T}}$ **then**
6:         $M_{t+1}, K_{t+1} \leftarrow$ retrain $M_t$ and $K_t$ with the pair loss function $L_{\rho,\lambda,\gamma}(M_t, K_t, D_t)$ as defined in equation 3         ▷ *periodical retrain*
7:     **end if**
8: **end for**
9: **Output**: $\underset{t}{\max}\, y_t$

---

## 4.2 COLLISION PENALTY

In this subsection, we aim to quantify the collision effect based on the definition proposed in Section 3.3. As illustrated in Figure 2, we feed pairs of data points into the neural network and obtain their latent space representations. Apart from maximizing the GP's likelihood, we concurrently calculate the amount of collision on each pair, and penalize only if the value is positive. For $x_i, x_j \in \mathcal{X}$, $y_i = f(x_i) + \epsilon$, $y_i = f(x_i) + \epsilon$ are the corresponding observations, and $z_i = g(x_i)$, $z_j = g(x_j)$ are the corresponding latent space representations.

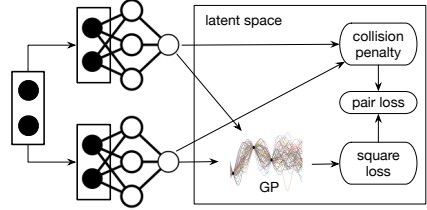

Figure 2: CoFLO schematic

---

[1]Note that we have introduced several hyper-parameters in the algorithm design; we will defer our discussion on the choice of these parameters to Section 5.

[2]To obtain an initial latent space representation, the labels do not have to be exact and could be collected from a related task of cheaper cost

We define the *collision penalty* as

$$p_{ij} = \max(\lambda|y_i - y_j| - |z_i - z_j|, 0) \tag{1}$$

where $\lambda$ is a penalty parameter that controls the smoothness of the target function $h : \mathcal{Z} \rightarrow \mathbb{R}$.

### 4.3 DYNAMIC WEIGHT

Note that it is challenging to universally eliminate the collision effect by minimizing the collision penalty and the GP's regression loss—this is particularly true with a limited amount of training data. Fortunately, in the optimization tasks it is often unnecessary to learn equally good representation for suboptimal regions. Therefore, we can dedicate more training resources to improve the learned latent space by focusing on the (potentially) near-optimal regions. Following this insight, we propose to use a weighted collision penalty function, which uses the objective values for each pair as importance weight in each iteration. Formally, for any pair $((x_j, z_j, y_j), (x_i, z_i, y_i))$ in a batch of observation pairs $D_t = \{((x_m, z_m, y_m), (x_n, z_n, y_n))\}_{m,n}$, we define the *importance-weighted penalty function* as

$$\tilde{p}_{ij} = p_{ij} w_{ij} \qquad \text{with} \qquad w_{ij} = \frac{e^{\gamma(y_i + y_j)}}{\sum\limits_{(m,n) \in D_t} e^{\gamma(y_m + y_n)}}. \tag{2}$$

Here the importance weight $\gamma$ is used to control the aggressiveness of the weighting strategy.

Combining the collision penalty and regression loss of GP, we define the *pair loss* function $L$ as

$$L_{\rho,\lambda,\gamma}(M_t, K_t, D_t) = \frac{1}{||D_t||^2} \sum_{i \in D_t, j \in D_t} (GP_{K_t}(M_t(x_i)) - y_i)^2 + (GP_{K_t}(M_t(x_j)) - y_j)^2 + \rho\tilde{p}_{ij}, \tag{3}$$

Here, $GP_{Kt}(M_t(x_i))$ denotes the Gaussian process's posterior mean on $x_i$ with kernel $K_t$ and neural network $M_t$ at timestep $t$. $\rho$ denotes the regularization weight; as we demonstrate in Section 5, in practice we often choose $\rho$ to keep the penalty at a order close to the regression loss.

### 4.4 THEORETICAL ANALYSIS

In this subsection, we provide a theoretical justification for the collision-free regularizer, by inspecting the effect of regularization on the regret bound of CoFLO. We first connect the proposed collision penalty in Equation 1 to Lipschitz-continuity, and then integrate it into the regret analysis to provide an improved regret bound.

**Lipschitz continuity of the target function** $h$   The collision penalty encourages the Lipschitz-continuity for $h$. Formally, the proposed regularization promotes to learn a latent space where $\forall x_i, x_j \in D, z_i = g(x_i), z_j = g(x_j) \in Z$,

$$|g(x_i) - g(x_j)| \le \lambda|f(x_i) - f(x_j)|$$

The above inquality reduces to the Lipschitz-continuity for $h$.  Unlike typical smoothness assumptions in GPs, a function can be non-smooth and still Lipschitz continuous.  Recently, Ahmed et al. (2019) leverage the Lipschitz-continuity of the objective function to propose improved acquisition functions based on the common acquisition functions, and provide an improved regret bound both theoretically and empirically. In the following, we show that running GP-UCB on the collision-free latent space amounts to an improvement in terms of its regret bound:

**Theorem 1.** *Let $\mathcal{Z} \subset [0, r]^d$ be compact and convex, $d \in N, r > 0, \lambda \ge 0$. Suppose that the objective function $h$ defined on $\mathcal{Z}$ is a sample from GP and is Lipschitz continuous with Lipschitz constant $\lambda$. Let $\delta \in (0, 1)$, and define $\beta_t = 2\log(\pi^2 t^2/6\delta) + 2d\log(\lambda r d t^2)$. Running the GP-UCB with $\beta_t$ for a sample $h$ of a GP with mean function zero and covariance function $k(x, x')$, we obtain a regret bound of $O^*(\sqrt{dT\gamma_T})$ with high probability.*

*Precisely, with $C_1 = 8/\log(1 + \sigma^{-2})$, we have*

$$\mathbb{P}\left[R_T \le \sqrt{C_1 T \beta_T \gamma_T} + 2\right] \ge 1 - \delta.$$

*Here $\gamma_T$ is the maximum information gain after $T$ iterations, defined as $\gamma_T := \max\limits_{A \subset Z, |A| = T} I(y_A; h_A)$.*

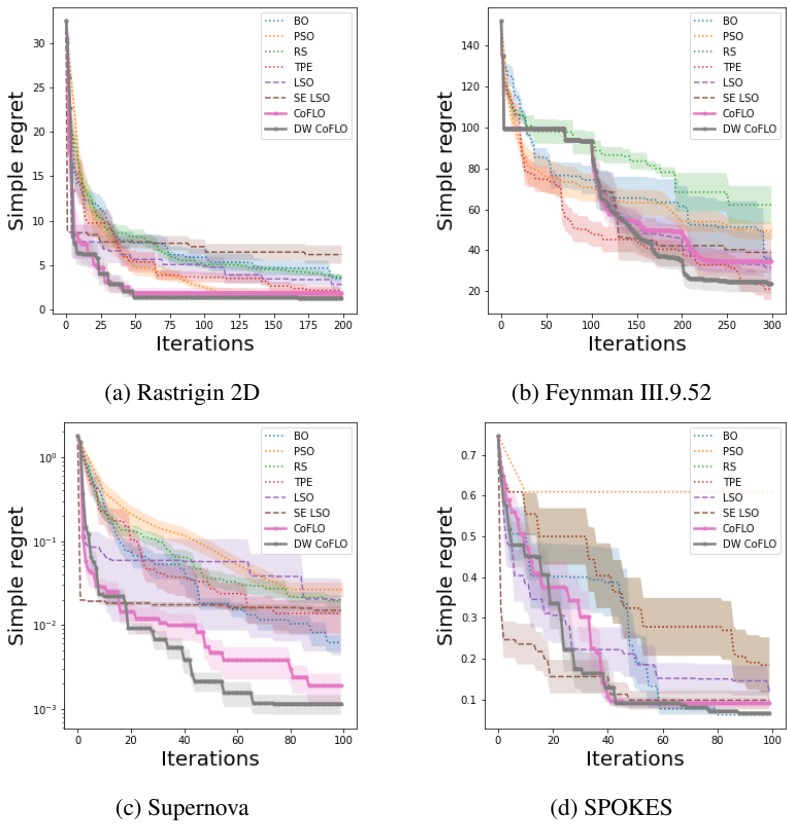

(a) Rastrigin 2D                    (b) Feynman III.9.52

(c) Supernova                       (d) SPOKES

Figure 3: Experiment results on four pre-collected datasets. Each experiment is repeated at least ten times. The colored area around the mean curve denotes the $\frac{\hat{\sigma}}{\sqrt{n}}$. Here $\hat{\sigma}$ denotes the empirical standard deviation. $n$ denotes the number of cases repeated in experiments. The hyper-parameters are set as the following. The retrain interval $\tilde{T}$ are set to be 100 iterations for 8c 8a and 8d, 200 for 8b. The Regularization parameters $\rho$ are set to be $1e^5$ for 8c 8a and 8d, $1e^3$ for 8b. The penalty parameter $\lambda$ are all set to be $1e^{-2}$. The weighting parameter $\gamma$ are set to be $1e^{-2}$. The prior mean $\mu_0$ are all set to be 0. The squared exponential kernel is used as the GP covariance for all the four experiments. We also demonstrate the median curves in the Appendix.

Comparing the above result to Theorem 2 of Srinivas et al. (2010) which offers the regret bound with the sub-gaussianity assumption on the objective functions derivative, the second part of our regret bound does not rely on $\delta$. The coefficients are also smaller as the deterministic bound on the derivative of $f$ avoids union bound.

**Remark.** *The collision penalty encourages $h$ to be Lipschitz-continuous on the latent space. Ideally, when the collision penalty $p_{i,j}(\lambda)$ term converges to zero for all data points in the latent space, we can claim that $h$ is Lipschitz-continuous with Lipschitz constant at most $\lambda$. Applying Theorem 1 with $\beta_t = 2\log(\pi^2 t^2/6\delta) + 2d\log(\lambda r dt^2)$, we can reduce the regret bound by choosing smaller $\lambda$. However, in practice, since the observation can be noisy, we need to choose a $\lambda$ big enough to tolerant the noise. A small $\lambda$ could make it difficult to learn a meaningful representation.*

## 5 EXPERIMENTS

In this section, we empirically evaluate our algorithm on two synthetic blackbox function optimization tasks and two real-world optimization problems.

## 5.1 EXPERIMENTAL SETUP

We consider four baselines in our experiments. The rudimentary random selection algorithm (RS) shows the task complexity. Three popular optimization algorithms, namely particle swarm optimization (PSO) (Miranda, 2018), Tree-structured Parzen Estimator Approach (TPE) (Bergstra et al., 2011), and standard Bayesian optimization (BO) (Nogueira, 2014) which uses Gaussian process as the statistical model and the upper confidence bound (UCB) as its acquisition function, are tuned in each task. Another baseline we consider is the sample-efficient LSO (SE LSO) algorithm, which is implemented based on the algorithm proposed by Tripp et al. (2020). We also compare the non-regularized latent space optimization (LSO), Collision-Free Latent Space Optimization (CoFLO) and the dynamically-weighted CoFLO (DW CoFLO) proposed in this paper. The performance for each task is measured on 10,000 pre-collected data points.

One crucial problem in practice is tuning the hyper-parameters. The hyper-parameters for GP are tuned for periodically retraining in the optimization process, by minimizing the loss function on a validation set. For all our tasks, we choose a simplistic neural network architecture $M$, due to limited and expensive access to labeled data under the BO setting. The coefficient $\rho$ is, in general, selected to guarantee a similar order for the collision penalty to GP loss. The $\lambda$ should be estimated according to the first several sampled data and tolerant the additive noise in the evaluation. $\gamma$ controls the aggressiveness of the importance weight. While $\gamma$ should not be too close to zero (which is equivalent to uniform weight), an extremely high value could make the regularization overly biased. Such a severe bias could possibly allow a heavily collided representation in most of the latent space and degrade regularization effectiveness. The value choice is similar to the inverse of the temperature parameter of softmax in deep learning Hinton et al. (2015). Here we use the first batch of observed samples to estimate the order of all observations and choose the appropriate $\gamma$.

## 5.2 DATASETS AND RESULTS

We now evaluate CoFLO on two synthetic datasets and two real-world datasets. In the experiments, all the input data points are mapped to a *one-dimensional latent space* by via the neural network. We demonstrated the improvement of CoFLO brought by the explicit collision mitigation in the lower-dimensional latent space in terms of average simple regret. We also include the median result and statistical test in the appendix.

**2D-Rastrigin** The Rastrigin function is a non-convex function used as a performance test problem for optimization algorithms. It was first proposed by RASTRIGIN (1974) and used as a popular benchmark dataset for evaluating Gaussian process regression algorithms (Cully et al., 2018). Formally, the 2D Rastrigin function is

$$f(x) = 10d + \sum_{i=1}^{d} x_i^2 - 10cos(2\pi x_i), \ d = 2$$

For convenience of comparison, we take the $-f(x)$ as the objective value to make the optimization tasks a maximization task. The neural network is pretrained on 100 data points. As illustrated by figure 8a , CoFLO and DW CoFLO could quickly reach the (near-) optimal region, while the baselines generally suffer a bigger simple regret even after an excessive number of iterations.

**Feynman III.9.52 Equation** Growing datasets have motivated pure data-backed analysis in physics. The dataset of 100 equations from the *Feynman Lecture on Physics* for the symbolic regression tasks in physics (Udrescu and Tegmark, 2020) could play the role as a test set for data-back analysis algorithms in physics. The III.9.52 we choose to test the optimization algorithms is

$$\rho_\gamma = \frac{p_d E_f t}{h/2\pi} \frac{sin((\omega - \omega_0)t/2)^2}{((\omega - \omega_0)t/2)^2}$$

The equations have 6 variables as inputs and are reported to require at least $10^3$ data for the regression task. The neural network is randomly initialized at the beginning. As illustrated by figure 8b, in the first 100 iterations, CoFLO and DW CoFLO behaves similarly to random selection. After the first training at iteration 100, CoFLO and DW CoFLO approach the optimum at a much faster pace compared to the baselines; among them, DW CoFLO shows a faster reduction in simple regret.

**Supernova** Our first real-world task is to perform maximum likelihood inference on 3 cosmological parameters, the Hubble constant $H_0 \in (60, 80)$, the dark matter fraction $\Omega_M \in (0, 1)$ and the dark energy fraction $\Omega_A \in (0, 1)$. The likelihood is given by the Roberson-Walker metric, which requires a one-dimensional numerical integration for each point in the dataset from Davis et al. (2007). The neural network is pretrained on one hundred data points. As illustrated by figure 8c, the simple regret SE LSO has a faster drop at the beginning, while later remained relatively stable and eventually ends at a similar level to LSO. These results demonstrate the efficiency of SE LSO when finding sub-optimal. However, without collision reduction, SE LSO couldn't outperform the LSO in the long run, where both reach their limitation. And the CoFLO and DW CoFLO demonstrate its robustness when close to the optimal as both constantly approach the optimal. Among them, DW CoFLO slightly outperform CoFLO.

**Redshift Distribution** The challenges in designing and optimizing cosmological experiments grow commensurately with their scale and complexity. Careful accounting of all the requirements and features of these experiments becomes increasingly necessary to achieve the goals of a given cosmic survey. SPOKES (SPectrOscopic KEn Simulation) is an end-to-end framework that can simulate all the operations and key decisions of a cosmic survey (Nord et al., 2016). It can be used for the design, optimization, and forecasting of any cosmic experiment. For example, some cosmic survey campaigns endeavor to observe populations of galaxies that exist at a specific range of redshifts (distances) from us. In this work, we use SPOKES to generate galaxies within a specified window of distances from Earth. We then minimize the Hausdorff distance between the desired redshift distribution and the simulation of specific cosmological surveys generated by SPOKES.

In our experiments, the neural network is pretrained on 200 data points. As illustrated by figure 8d, the simple regret of SE LSO drops faster at the initial phase. However, when it gets close to the (near-) optimal region where simple regret is approximately 0.15, it is caught up by both CoFLO and DW CoFLO, and eventually gets slightly outperformed. Such a result indicates that the collision problem could have more impact when the algorithm gets close to the optimal region. Notice that the rudimentary BO eventually outperformed the non-regularized LSO, indicate that without mitigation of collision, the learned representation could worsen the performance in the later stage when the algorithm gets close to the optimal. In conclusion, the mitigation of collision like CoFLO is necessary to further improve the later performance of LSO, when collision matters more in the near-optimal areas.

## 5.3 DISCUSSION

In general, our experimental results consistently demonstrate the robustness of our methods against collisions in the learned latent space. Our method outperforms all baselines; when compared to the sample-efficient LSO, the dynamically-weighted LSO performs better in most cases and shows a steady capability to reach the optimum by explicitly mitigating the collision in the latent space. In contrast, the Sample-efficient LSO might fail due to the collision problem.

## 6 CONCLUSION

We have proposed a novel regularization scheme for latent space based Bayesian optimization. Our algorithm—namely CoFLO—addresses the collision problem induced by dimensionality reduction, and improves the performance for latent space-based optimization algorithms. The regularization is proved to be effective in mitigating the collision problem in learned latent space, and therefore can boost the performance of the Bayesian optimization in the latent space. We demonstrate strong empirical results for CoFLO on several synthetic and real-world datasets, and show that CoFLO is capable of dealing with high-dimensional input that could be highly valuable for real-world experiment design tasks such as cosmological survey scheduling.

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

## A  REGRET BOUND FOR A LIPSCHITZ-CONTINUOUS OBJECTIVE FUNCTION

In this section, we provide the detailed proof for Theorem 1. We first modify Lemma 5.7 and Lemma 5.8 in Srinivas et al. (2010) since we are assuming the deterministic Lipschitz-continuity for $h$. Use the same analysis tool $Z_t$ defined as a set of discretization $Z_t \subset Z$ where $Z_t$ will be used at time $t$ in the analysis.

We choose a discretization $Z_t$ of size $(\tau_t)^d$. so that $\forall z \in Z$,

$$||z - [z]_t||_1 \leq rd/\tau_t \tag{4}$$

where $[z]_t$ denotes the closest point in $Z_t$ to $z$.

**Lemma 1.** *Pick $\delta \in (0,1)$ and set $\beta = 2log(\pi_t/\delta) + 2dlog(Lrdt^2)$, where $\sum_{t \geq 1} \pi_t^{-1} = 1$, $\pi_t > 0$. Let $\tau_t = Lrdt^2$. Hence then*

$$|h(z^*) - \mu_{t-1}([z^*]_t)| \leq \beta_t^{1/2}\sigma_{t-1}([z^*]_t) + 1/t^2 \quad \forall t \geq 1$$

*holds with probability $\geq 1 - \delta$. Here $z^* := g(x^*)$.*

*Proof.* Using the Lipschitz-continuity and equation 4, we have that

$$\forall z \in Z, |h(z) - h([z]_t)| \leq Lrd/\tau_t$$

By choosing $\tau_t = Lrdt^2$, we have $|Z_t| = (Lrdt^2)^d$ and

$$\forall z \in Z, |h(z) - h([z]_t)| \leq 1/t^2$$

Then using Lemma 5.6 in Srinivas et al. (2010), we reach the expected result. ☐

Base on Lemma 5.5 in Srinivas et al. (2010) and Lemma 1, we could have the following result directly.

**Lemma 2.** *Pick $\delta \in (0,1)$ and set $\beta = 2log(2\pi_t/\delta) + 2dlog(Lrdt^2)$, where $\sum_{t \geq 1} \pi_t^{-1} = 1$, $\pi_t > 0$. Then with probability $\geq 1 - \delta$, for all $t \in N$, the regret is bounded as follows:*

$$r_t \leq 2\beta_t^{1/2}\sigma_{t-1}(z_t) + 1/t^2$$

*Proof.* Using the union bound of $\delta/2$ in both Lemma 5.5 in Srinivas et al. (2010) and Lemma 1, we have that with probability $1 - \delta$:

$$
\begin{aligned}
r_t &= h(z^*) - h(z_t) \\
&\leq \beta_t^{1/2} \sigma_{t-1}(z_t) + 1/t^2 + \mu_{t-1}(z_t) - h(z_t) \\
&\leq 2\beta_t^{1/2} \sigma_{t-1}(z_t) + 1/t^2
\end{aligned}
$$

which complete the proof. $\qquad\square$

Now we are ready to use the Lemma5.4 in Srinivas et al. (2010) and Lemma 2 to complete the proof of Theorem 1.

*Proof.* Using Lemma5.4 in Srinivas et al. (2010), we have that with probability $\geq 1 - \delta$:

$$
\sum_{t=1}^{T} 4\beta_t \sigma_{t-1}^2(x_t) \leq C_1 \beta_T \gamma_T \quad \forall T \geq 1
$$

By Cauchy-Schwarz:

$$
\sum_{t=1}^{T} 2\beta_t^{1/2} \sigma_{t-1}(x_t) \leq \sqrt{C_1 \beta_T \gamma_T} \quad \forall T \geq 1
$$

Finally, substitute $\pi_t$ with $\pi^2 t^2/6$ (since $\sum 1/t^2 = \pi^2/6$). Theorem 1 follows. $\qquad\square$

## B   VISUALIZATION OF THE COLLISION EFFECT IN LATENT SPACE

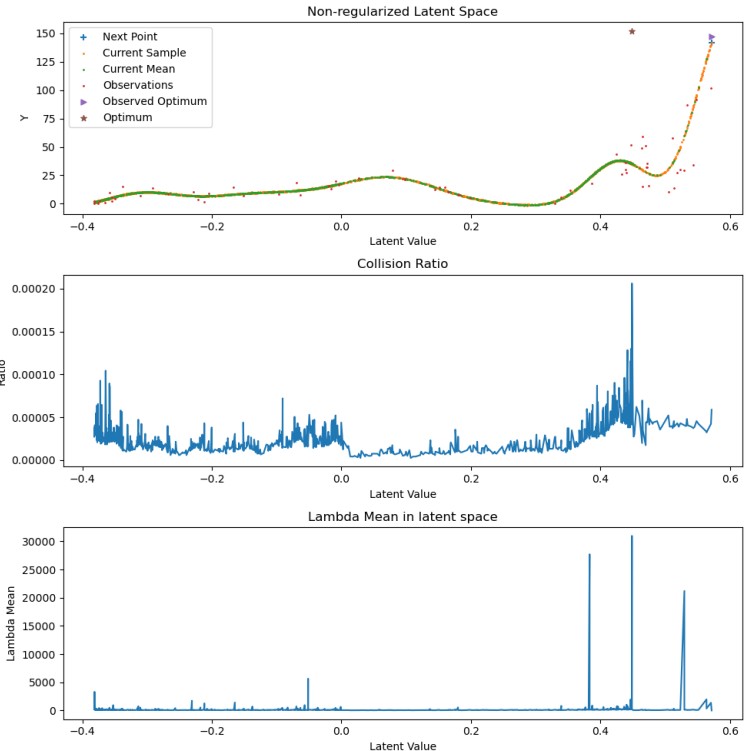

Figure 4: Illustrate the collision and quantified measurement of the collision. Here we propose two quantity measurement of the collision. For the second graph the y axis of the is the ratio of exceeding $|y_1 - y_2| > |z_1 - z_2| * L$. And for the third graph, the y axis of the third column is the mean of $\lambda = |y_1 - y_2|/|z_1 - z_2|$.

We demonstrate the collision effect in the latent space. We trained the same neural network on Feynman dataset with 101 data points which demonstrate the latent space after two retrains with the retrain interval set to be 50 data points. The regularized one employed DW CoFLO, with the regularization parameter $\rho = 1e^5$, penalty parameter $\lambda = 1e^{-2}$, retrain interval $\tilde{T}$, weighting parameter $\gamma = 1e^{-2}$ and the base kernel set to be square exponential kernel. The non-regularized one employed LSO.

## C   SUPPLEMENTAL MATERIALS ON ALGORITHMIC DETAILS

### C.1   ALGORITHMIC DETAILS ON NEURAL NETWORK ARCHITECTURE

As the main goal of our paper was to showcase the performance of a novel collision-free regularizer, we picked our network architectures to be basic multi-layer dense neural network:

For SPOKES, we used a 5-layer dense neural network. Its hidden layers consist of 16 neurons with Leaky Relu nonlinearities, 8 neurons with Sigmoid nonlinearities, 4 neurons with Sigmoid nonlinearities, and 2 neurons with Sigmoid nonlinearities respectively. Each hidden layer also applies a 0.2 dropout rate. The output layer applies Leaky Relu nonlinearity.

For SuperNova, Feynman, and Rastrigin 2D, we used a 4-layer dense neural network. Its hidden layers consist of 8 neurons with Sigmoid nonlinearities, 4 neurons with Leaky Relu nonlinearities, and 2 neurons with Leaky Relu nonlinearities respectively. Each hidden layer applies a 0.2 dropout rate. The output layer also applies Leaky Relu nonlinearity.

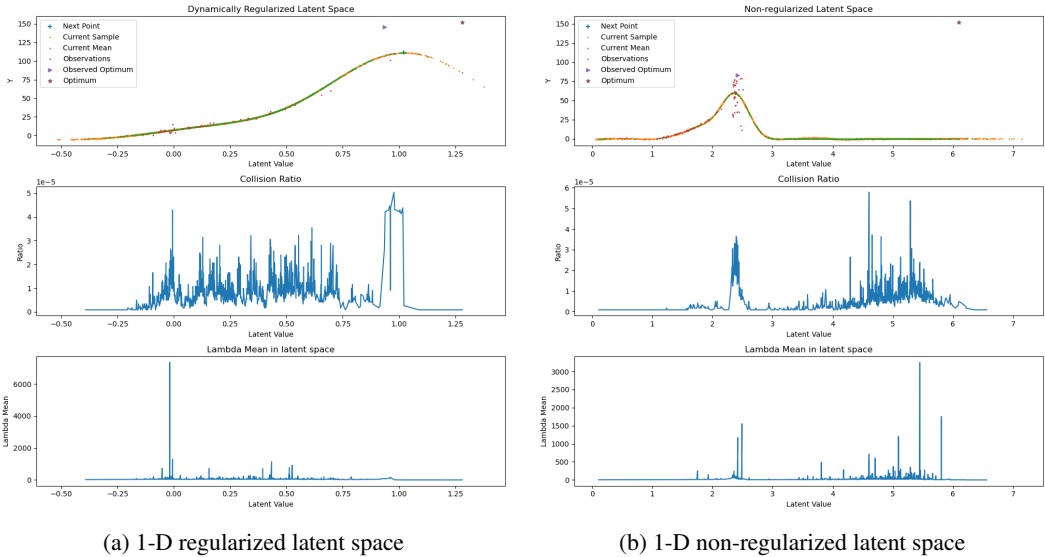

(a) 1-D regularized latent space

(b) 1-D non-regularized latent space

Figure 5: Illustrate the 1-D latent space of Feynman III.9.52 dataset. 5a shows a regularized latent space with a few observable collisions. 5b shows a non-regularized latent space with bumps of collisions especially around the maxima among the observed data points. Besides, having fewer collisions in the latent space contribute to the optimization through improving the learned Gaussian process. We observed in this comparison that the next point selected by the acquisition function of the regularized version is approaching the global optima, while the next point in the non-regularized version is trying to solve the uncertainty brought by the severe collision near the currently observed maxima.

The neural networks are trained using ADAM with a learning rate of $1e^{-2}$.

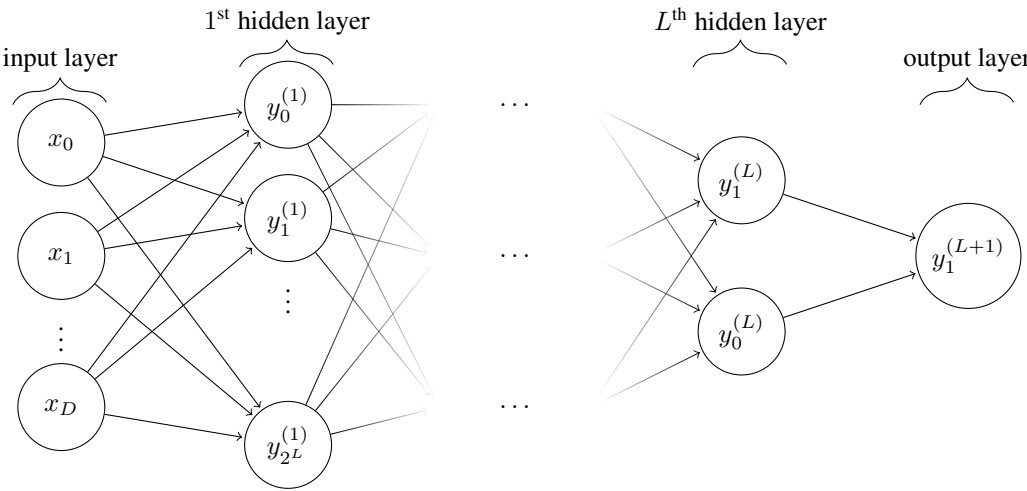

Figure 6: Network graph of a $(L+1)$-layer dense network with $D$ input units and 1 output units. In our experiments L is set to be 4 for Rastrigin 2D, Feynman II.9.52, Supernova, and 5 for SPOKES.

## C.2 PARAMETER CHOICES

We further investigate the robustness of parameter choices of both the regularization parameter $\rho$ and the penalty parameter $\lambda$ on SPOKES dataset. We show the result in the figures below.

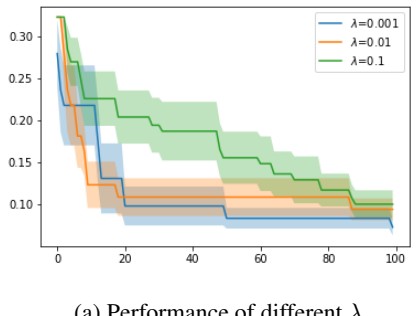 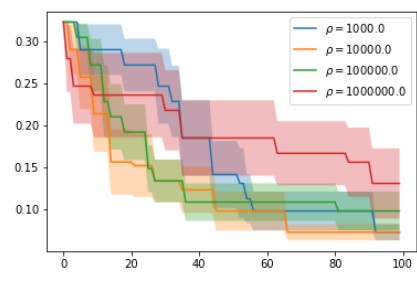

(a) Performance of different $\lambda$        (b) Performance of different $\rho$

Figure 7: Simple regrets under different parameter settings on the SPOKES dataset. 7b shows that a regularization parameter too big could distract the training process and downgrade the performance. And we choose $\rho = 1e^5$ in practice as it maintains the collision penalty in the same order of the regression loss of GP in equation 3. 7a shows that a relatively small value could do a good job. We believe that's because the wide range of objective values of the tested dataset needs to be mitigated. The curves demonstrate the decent performance of CoFLO as long as the parameters are not set to be too large.

# D   ADDITIONAL EXPERIMENTAL RESULTS

We added both the detailed median curves and the p-values of the Welch's t-tests of the experiments we discussed in section 5.

## D.1   MEDIAN CURVE

The median curves demonstrate similar trends to the mean curves. In the four experiments, DW CoFLO consistently demonstrates a superior performance over the baselines.

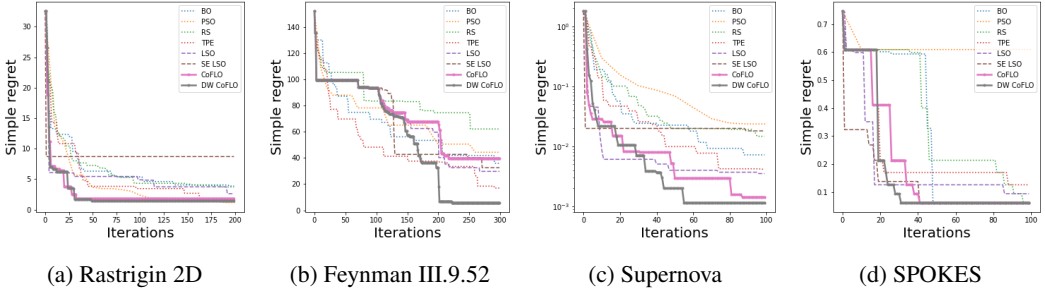

(a) Rastrigin 2D      (b) Feynman III.9.52      (c) Supernova      (d) SPOKES

## D.2   P-VALUES

The table shows the p-values of Welch's t-tests of the experiments. It demonstrates the significance of the improvement brought by DW CoFLO over the baselines.

| Data | BO | RO | TPE | LSO | SE-LSO | CoFLO |
|------|-----|-----|-----|-----|--------|-------|
| Rastrigin-2D | $1.07e^{-4}$ | $3.88e^{-8}$ | $1.01e^{-2}$ | $6.38e^{-3}$ | $1.10e^{-5}$ | $4.23e^{-1}$ |
| Supernova | $3.24e^{-3}$ | $3.61e^{-3}$ | $3.18e^{-2}$ | $3.43e^{-1}$ | $1.41e^{-8}$ | $2.62e^{-1}$ |
| Feynman | $1.73e^{-1}$ | $1.52e^{-07}$ | $8.20e^{-1}$ | $288e^{-1}$ | $6.37e^{-1}$ | $2.25e^{-1}$ |
| SPOKES | $4.62e^{-1}$ | $9.90e^{-3}$ | $2.64e^{-1}$ | $4.17e^{-2}$ | $2.87e^{-3}$ | $4.11e^{-1}$ |

