# OpenReview forum: "Learning Collision-free Latent Space for Bayesian Optimization"
_ICLR.cc/2021/Conference — Reject_

### Official Review · AnonReviewer2 · 2020-10-24
**Official Blind Review #2**

**Rating:** 5
**Confidence:** 4

**Review:**

The paper proposes (1) a new regularization strategy for the latent space-based BO, (2) an optimization aware dynamic weighting for adjusting the collison penalty to improve BO, (3) theoretical analysis for the BO on the latent space. The idea for the regularization is to take in pairs of data points and penalizes those too close in the latent space compared to their target space distance

The paper makes an interesting observation that the learned representation (for BO to deal with complex object or high-dimension) often leads to collision in the latent space: two points with significant different observations get too close in the learned latent space. Collisions could be regarded as additional noise introduced by the traditional neural network, leading to degraded optimization performance

The mapping by neural network to learn g: D->Z is typically considered as the regression problem in which the neural network should learn the property that similar input should have similar output.

A pair loss is integrated in learning the neural network. The dynamic weight improves the learned latent space by focusing on the potential high-value region.

The idea of using constraint in the latent space has also been studied in [1].

Despite of the good motivation, the paper execution is not yet demonstrated the effectiveness of the proposed approach for three main reasons:

(1) The experiments using 4 settings are quite simple and havenot yet satistisfactorily convinced why the proposed approach performs intuitively better. It can be improved further by demonstrating the collison-effect in more challenging task, such as automatic chemical design [2].

(2) The theoretical analysis follows and extends from Srinivas et al 2010.

(3) Fig 1 demonstrates the collision in 1d using the non-regularized latent space? It will be useful if you can add another figure in the same setting using the regularized latent space.

The writing and presentation can be improved more.


Typo:
Section 5.1: “promotse"
Remark: why “Choosing” is capitalized in the middle of the sentence?
“UCB use the upper…” => “UCB uses the upper….”

[1] Kusner, M. J., Paige, B., & Hernández-Lobato, J. M. (2017, June). Grammar Variational Autoencoder. In Proceedings of the 34 th International Conference on Machine Learning, Sydney, Australia, PMLR 70, 2017 (Vol. 70, pp. 1945-1954). ACM.

[2] Griffiths, Ryan-Rhys, and José Miguel Hernández-Lobato. "Constrained Bayesian optimization for automatic chemical design using variational autoencoders." Chemical science 11.2 (2020): 577-586.

---

> ### Author Response · Authors · 2020-11-25
> **Authors' response to AnnoReviewer2 (on theoretical analysis and additional experiments)**
>
> Thank you for the detailed review and suggestions!
>
> We will fix the editorial issues in our revision. We also appreciate your suggestion in adding an illustration of the regularized latent space---we will incorporate the changes in the revision.
>
> Since the questions on the theoretical analysis and empirical results are common to other reviewers, please kindly find our responses following the pointers below.
>
> - **[Justification of the theoretical results]**
> Please refer to our response to [Q1 in of AnnoReviewer3](https://openreview.net/forum?id=bGZtz5-Cmkz&noteId=bIc06LC0t3d).
>
> - **[Demonstration of the collision effect with complex inputs]**
> Please refer to our response to [Q6 in of AnnoReviewer3](https://openreview.net/forum?id=bGZtz5-Cmkz&noteId=bIc06LC0t3d).

---

### Official Review · AnonReviewer4 · 2020-10-28
**The paper has some contributions, however, it is not ready for publish yet.**

**Rating:** 3
**Confidence:** 3

**Review:**

The paper proposes a method to avoid collision for the latent space based Bayesian optimization method. The main idea is to add a regularization term into the training process. Theoretical analysis is also conducted to understand the performance of the proposed method.

Although the idea of the proposed method is somewhat interesting, I do have many concerns for the paper.

1) The writing is not good, so it makes it hard to understand the work. In particular, the English of the paper is frequently bad (wrong grammar, typos, unfinished sentences). The maths notations are occasionally not consistent. For example sometimes, the penalty is defined as p_[i,j}, sometimes it is denoted as p_{ij}.
2) Section 4.2 is too ambiguous. What are z_i, z_j in the equation in Section 4.2? Based on the notation of the latent space Z, I can guess z_i, z_j are the values in the latent space, but this should be clearly mentioned in the paper. Also, what does \lambda represent? And how to set it in practice? I went through the 2nd paragraph in Section 6.2 and still feel unclear how to set this hyperparameter in practice.
3) Section 4.3 is also not clear. What is the intuition behind the weight \omega_{ij}? What do \gamma and \rho represent? How to set them? And what does GP_{Kt}(M_t(x_i)) (in Eq. (1)) denote?
4) Regarding the theoretical analysis, unless I miss something, it is just the standard theorem as in Srinivas et al. (2010), but replace the assumption of the objective function f being a sample path from the GP, by the assumption of the latent space function h being a sample path from the GP? In which cases this assumption is satisfied? And what does it mean by “comparing to Theorem 2 in Srinivas et al. (2010), the second part of the regret bound doesn’t rely on \delta"? As much as I understand, the regret bound in Theorem 1 is the same as the one in Theorem 2 in Srinivas et al. (2010).
5) Regarding the experiments, the experiments are only conducted on low-dimensional problems (2D, 6D, 3D, …), which is contradict with the motivation of the work (BO for high dimensional inputs). Besides, what does it mean when the neural networks are pretrained on a number of data points? Do we know the corresponding function values of these data points in advance? If yes, for the baseline methods the paper compares with, are these data points employed in these baseline optimization procedures?

---

> ### Author Response · Authors · 2020-11-25
> **Authors' response to AnnoReviewer4 (Clarifications to questions on editorial issues, algorithmic details, analysis, and experiments)**
>
> Thank you for the detailed review and comments. Below please find our responses (**A**) to the main questions (**Q**).
>
> **Q1 [Clarifications to Section 4]**
>
> **A**: Thanks for pointing out the notation issues and the ambiguities in writing. We appreciate the detailed feedback, and will fix the editorial issues in the revised version.
>
> We clarify the meaning of several key parameters below:
>
> * For $x_i, x_j$, we denote $z_i = g(x_i)$ and $z_j = g(x_j)$.
>
> * $\lambda$ is the regularization coefficient, which is used to balance the regularizer and the regression loss (i.e. induced by the neural network and kernel learning).
>
> * $\rho$ controls the smoothness of the latent space. We estimated the proper value by initial tries of the neural network and kernel learning on the datasets. We are also adding a study of the $\rho$ choice.
>
> * $\gamma$ controls the aggressiveness of the dynamic weighting. The value choice is similar to the inverse of the temperature parameter of softmax in deep learning.
>
> * $GP_{Kt}(M_t(x_i))$ in Eq. (1) denotes the gaussian process's posterior mean on $x_i$ with kernel $K_t$ and neural network $M_t$ at timestep $t$.
>
>
> For the common questions/concerns on the significance of our theoretical results, experimental setup, and other algorithmic details, please refer to our response below:
>
> - **[Justification of the theoretical results]**
> Please refer to our response to [Q1 of AnnoReviewer3](https://openreview.net/forum?id=bGZtz5-Cmkz&noteId=bIc06LC0t3d).
>
> - **[Pre-train & Budget]**
> Please refer to our response to [Q2 of AnnoReviewer3](https://openreview.net/forum?id=bGZtz5-Cmkz&noteId=bIc06LC0t3d).
>
> - **[Clarifications on the parameter setting and baselines]**
> Please refer to our response to [Q3 of AnnoReviewer3](https://openreview.net/forum?id=bGZtz5-Cmkz&noteId=bIc06LC0t3d).

---

### Official Review · AnonReviewer3 · 2020-10-28
**could be a nice paper if applied on the right problems and demonstrated clearly**

**Rating:** 4
**Confidence:** 4

**Review:**

This paper proposes a regularization technique in training a latent variable model so that points with different functions are pushed apart. It’s demonstrated that the proposed technique can boost regret bound and empirical performance.

Overall, I think it’s a nice paper, but I don’t think the current presentation is good enough for publication at ICLR.

comments & questions:

1. It’s a natural idea to add a Lipchitz-like regularization loss to mitigate “collision”.  The theoretical result seems a straightforward derivative of the Srinivas et al. (2010), but I don’t really see the novelty of the theoretical result, since Lipchitz continuity is implicitly determined by the kernel function?

2. the proposed method needs to pretrain the neural network with 100 or 200 points. It’s not clear to me what it means by “pre-train”. Is it supervised or unsupervised? Which 100 points are chosen for pretraining? if it’s supervised, did you count them in the optimization budget? that means if you pre-train on 100 labeled points, then perform 100 BO iterations, a fair comparison to standard BO would grant it a budget of 200 function evaluations.

3. there are several parameters, such as $\lambda$, $\gamma$, how are they chosen exactly? how sensitive are these parameters?
What exactly is the “standard BO” algorithm from Nogueira (2014)? Is it UCB? EI?

4. Seems all the benchmark functions have continuous domain with already low dimensions, e.g., the Rastrigin 2D only has 2 dimensions. Do you further reduce the dimension to one with the neural network? It would be great if you could plot the function on latent space. Same for other benchmarks, since they are not very high dimensional.

5. From the experiments, I don’t really see if it’s true that the baselines lose because they have collision problems. Is it possible to design some experiment to demonstrate that?

6. To me seems the work could be more motivated by input domains such as graphs or other discrete structures, at least for the benchmarks in the experiments I don’t see why they need this method despite the claimed superior performance. For your reference, some notable work on Bayesian optimization in latent space for discrete objects:
* Kusner et al. (ICML 2017), grammar VAE
* Jin et al. (ICML 2018), JT-VAE
* Zhang et al. (NeurIPS 2019), D-VAE
* ...

Minor:
the formula for posterior GP mean and covariance assumes zero prior mean, which was not explicitly pointed out.
in 3.1, most popular acquisition should definitely include expected improvement

there are many typos:
in Abstract: significant different -> significantly different
in Related Work: taskss -> tasks
in Related Work third paragraph: smooth(of… -> add space (and many other places)
in 3.1: ”wiggles” first quote wrong direction
In 3.1: “the acquisition function $\alpha$ … use it -> an acquisition function … uses it
in 3.1: “then use the sample as the acquisition function …, need to add period
in 4.1: base on  -> based on
in 5.1: “promotse” -> promotes?
...

---

> ### Author Response · Authors · 2020-11-24
> **Authors' response to AnnoReviewer3 (Clarifications on the significance, algorithmic details, results, and motivation)**
>
> Thank you for the detailed review and comments. Below please find our responses (**A**) to the main questions (**Q1-6**).
>
>
> **Q1 [Justification of our theoretical results]**
> *(...I don’t really see the novelty of the theoretical result, since Lipchitz continuity is implicitly determined by the kernel function?...)*
>
> **A**: Section 5.2 are mainly used as a theoretical motivation for introducing the regularizer. As rightfully pointed out by the reviewer, the regret bound could be viewed as a corollary of the regret bound of Srinivas et al. (2010). In response to your concern, we will rephrase our description of the theoretical result to clarify the novelty and positioning of our work in terms of theoretical contribution.
>
> As is mentioned in Srinivas et al. (2010) proof, the Lipchitz continuity is not deterministically constrained by the kernel function. We are hoping to show that explicit constrain of Lipchitz continuity has both practical and theoretical significance in terms of regret. More specifically, Srinivas et al. use Lipchitz continuity as a standard assumption; however, as we demonstrated in our case study (Fig. 1), **such assumptions do not necessaraliry hold in practice**, especially when the off-the-shelf kernel function does not well capture the covarience structure. Hence, in this paper, we explicitly encourage the *learned kernel* to exhibit such a property.
>
>
> **Q2 [Pre-train & Budget]**
> *(...the proposed method needs to pretrain the neural network...)*
>
> **A**: We follow a typical two-phase data collection workflow in our experimental setup: First, in the pre-training phase, a broad dataset (e.g. from the same or similar data domain) is collected to learn (generic) representations; second, a (denser) dataset is collected for the target task, and the (generic) representation is adapted.
>
> In our experiments, the pre-training phase is supervised and is used to train an initial neural network and the kernel function. In practice, it is reasonable to pre-train the neural network with related tasks. We used randomly selected data points with low objective values for pre-training (e.g. similar to using a pretrained CNN on ImageNet for image classification), which is later excluded in the optimization phase dataset.
>
> Since the pre-training phase is used as an initialization process rather than part of the optimization, we excluded it from the budget for sequential optimizaiton.
>
>
> **Q3 [Clarifications on the parameter setting and baselines]**
>
> **A**: The parameters are chosen based on parameter sweep during initial runs of the algorithm. The general principle, as detailed in Section 6.1, is to choose the coefficient ρ such that the pair loss is of the same order as the collision penalty; $\lambda$ and $\gamma$ are cross-validated to avoid extreme scenarios (e.g. either too aggressive or conservative reweighting). The effect of regularizor is reasonably robust to small changes of the hyperparameters, as long the above principle is satisfied.
>
> By "standard BO" we refer to UCB algorithm---we will clarify this in our revision.
>
>
> **Q4 [Visualization of the Latent Space]**
> *(...It would be great if you could plot the function on latent space...)*
>
> **A**: Thanks for the suggestion! We are using 1-d latent space in all four experiments when regularizing the neural network.
>
> We will include a side-by-side comparison of the learned latent spaces---when trained with/without the regularizer---to clearly demonstrate the regularizer's effect. We will add the additional plot to the revision.
>
>
> **Q5 [Effect of the regularizer]**
> *(...I don’t really see the baselines lose because ...collision problems...)*
>
> **A**: Please note that LSO could be viewed as a controlled baseline for demonstrating the collision effect, since the only difference between LSO and CoFLO is that LSO does not use regularization. The sharp contrast between LSO and CoFLO on the Rastrigin 2D, Supernova, and SPOKES datasets demonstrated the LSO loses because of the collision problems.
>
>
> **Q6 [Structured inputs]**
>
> **A**: Thanks for pointing out the notable works on Bayesian optimization for various high-dimensional inputs. We acknowledge that it would be more comprehensive and convincing to include discussions on datasets with structured inputs. However, we are hoping to demonstrate that collision in the latent space, even when the original input space is low-dimensional and unstructured, could be utilited to design better BO algorithms.
>
> The collision could be caused by information loss in dimension reduction. For a complicated neural network, we wouldn't expect a huge loss of information in the forward propagation; yet in budgeted optimization tasks such as Bayesian experimental design, it is often challenging to obtain adequate data points during the optimization process to train such a deep neural network. Thus we are aiming at mitigating the collision problem when information loss is hard to avoid even with traditional approaches.

---

### Official Review · AnonReviewer1 · 2020-10-29
**Official Blind Review #1**

**Rating:** 4
**Confidence:** 4

**Review:**



# summary

This paper proposed a method that can penalize collisions in latent space. To
be more concrete, for a model which combines a neural network and a Gaussian
Process, e.g. deep kernel learning, the learned latent features for two
different inputs can be very close. In the following GP modeling, these two
similar latent features will cause difficulties since the covariance between
them will be large although these two inputs are quite different.


# cons

1.  The general idea presented in this work is very interesting. This is also a
    very realistic treatment, i.e. collisions in the latent space.
2.  Although not a new technique, e.g. siamese network, triplet loss, I like
    the idea of penalizing close points in latent space combined with a GP. It
    implicitly incorporate prior knowledge in modeling the GP, which usually
    boost the performance of a GP.


# pros

1.  I am doubtful about the correctness of eq(1). Without a treatment of
    stochastic variational inference, the marginal likelihood of GP cannot be
    factorized into a product of per data points. This means the batched update
    of GP in eq(1) will not produce a correct GP model, if I understand eq(1)
    correctly. Can the authors explain this batched update?
2.  I think experimental results should be extended to include a comparison
    with SMAC and TPE, which are two strong baselines. Although this work focus
    on GP based BO, empirical results of SMAC and TPE **without** considering
    collisions will make this work more convincing.
3.  The experimental settings used in this work are not detailed, e.g. how many
    units in each layer in the neural network, etc. Empirical results are also not sufficient.
4. In Figure3, the line is the mean curve instead of median of at least 10 experiments. However, without a statistical test, it is hard to tell whether the proposed approach is better than other competing methods.


# questions

1.  It is not clear to me why the retrain interval $\tilde{T}$ is set to be 100
    for 3c, 3a and 3d in Figure 3. In Algorithm 1, the latent model is updated
    every $\tilde{T}$ iterations. In Figure 3, the total iteration numbers for
    3c, 3a and 3d are 100, 200 and 100 respectively. This means for 3c and 3d,
    the latent model is updated only once and this update happens at the end of
    BO. After the update, the model will never be used. Can the authors comment
    on this?

Overall speaking, I am afraid this paper doesn't contain necessary details and
the theoretical results are not strong enough.

---

> ### Author Response · Authors · 2020-11-24
> **Authors' response to AnnoReviewer1 (Clarifications on the model udpate rule, experimental details, and additional baseline)**
>
> Thank you for the detailed review and comments. Below please find our responses (**A**) to the main questions (**Q1-4**).
>
> **Q1: [Justification of the kernel/model updates]**
> *(“correctness of eq(1)”)*
>
> **A**: Please note that Eq. (1) specifies the loss function for training a (deep) kernel, which is *later* used for Gaussian process regression. This aligns with the learning-based Gaussian process regression literature (e.g. the deep kernel learning framework by Wilson et al. (2015)).
>
> The batched update is for optimizing the parameters for the deep kernel (i.e., both the neural network weights and the base kernel parameters). Given the data representation and the kernel parameters learned via Eq. (1), we then calculate the Gaussian process posterior according to the Bayes rule specified in Section 3.1.
>
> Therefore, our algorithmic procedure does not require an explicit variational inference module to output a proper GP.
>
> Does this clarify your question regarding the correctness of Eq. (1)?
>
> **Q2: [Experimental Comparison]**
> *(I think experimental results should be extended to include a comparison with SMAC and TPE)*
>
> **A**: Thanks for the suggestions regarding additional baselines and detailing the experimental setup!
>
> In the context of Bayesian optimization, SMAC subsumes the BO algorithm, which we already included in our existing baselines. Additionally, we have evaluated DW CoFLo against TPE on the benchmark datasets mentioned in Section 6.2.
>
> As a preview of the new results, we have observed on the Feynman dataset, that DW-CoFLO consistently outperforms the TPE baseline by a large margin when exhausting the budget of 300 iterations. Furthermore, we observe that the performance of TPE has a significantly higher variance than the baselines across the four datasets considered in this paper. We summarize the performance comparison between TPE and CoFLO in the table below:
>
> |    | SuperNova | Feynman | SPOKES  | Rastrigin 2d|
> |----------|--------------|-------------|-------------|-------------|
> |TPE | -0.795| 73.26| -4.11 | -21.13|
> |DW-CoFLO| -0.163| 118.8| -4.08 | -15.11|
>
> We will incorporate these additional experimental results in the revision.
>
> **Q3: [Experiment Setting Details]**
> *("The experimental settings used in this work are not detailed")*
>
> **A**: In the following, we provide a detailed description of our experimental setup---including the neural networks used, parameter configuration, etc.
>
> The main goal of our paper was to showcase the performance of a novel collision-free regularizer. We picked our network architectures to be basic multi-layer dense neural network:
>
> For SPOKES, we used a 5-layer dense neural network. Its hidden layers consist of 16 neurons with Leaky Relu nonlinearities, 8 neurons with Sigmoid nonlinearities, 4 neurons with Sigmoid nonlinearities, and 2 neurons with Sigmoid nonlinearities respectively. Each hidden layer also applies a 0.2 dropout rate. The output layer applies Leaky Relu nonlinearity. For SuperNova, Feynman, and Rastrigin 2D, we used a 4-layer dense neural network. The difference between it and the 5-layer neural network is that we remove the first hidden layer of 16 neurons. The neural networks are trained using ADAM with a learning rate of 1e-2.
>
> We will include an illustration of the network architecture and add the detailed experimental configurations in the revision.
>
> **Q4: [Experiment Result]**
> *(the line is the mean curve instead of median...without a statistical test, it is hard to tell whether the proposed approach is better...)*.
>
> **A**: We have observed that the median curves follow the same trends as the mean---we will provide the detailed median plots in the revised appendix. In addition, we compared the regret distribution of each baseline approach against the regret distribution of DW-CoFLO, under the same cut-off of the optimization budget. The table shows a preview of the p-values of the Welch's t-tests on the Rastrigin 2d dataset:
>
> |    | BO | RO | TPE | LSO  | SE-LSO| CoFLO
> |----------|--------------|-------------|-------------|-------------|-------------|-------------|
> |p-value | 1.07e-3| 3.87e-8| 1.00e-2| 6.37e-2| 1.10e-5| 4.23e-1|
>
> Does this address your concern over the illusion of the experiment results?
>
> **Q5: [Retrain interval]**
> *("It is not clear to me why the retrain interval is set to be 100...")*
>
> **A**: For interval problems in 3c, 3d, we regarded the first 100 points as a pure pre-train of the kernel and neural network rather than a part of the Bayesian optimization process. In practice, it is common to pre-train the neural network and kernel with a relative dataset. And here, we selected data points with the lowest objective value for the pre-train phase. Thus the first 100 points results are excluded from the performance comparison. We will rephrase the 3c, 3d experiment result's elaboration to be the performance after 100 points pre-train of the kernel and neural network.

---

### Author Response · Authors · 2020-11-25
**Summary of changes in rebuttal revision**


We thank the reviewers for their detailed comments and valuable suggestions.  We studied the reviews and discussions carefully and modified our paper accordingly. Our revision followed the same list of actions proposed in our rebuttal response and further feedback from the reviewers.

Next, we summarize the (key) changes included in our revision for each paper section.

**[1. Introduction]**

We have clarified and highlighted our contribution in the revised introduction section. We also clarified the notations used in the method section. We conducted the additional comparisons between CoFLO and TPE, with median curves and statistical tests added to the appendix to further demonstrate the significance of our experiment results. The elaboration of experiment settings is also revised with more details added to the appendix.

**[2. Related work]**

We further clarified the connections between the proposed work and Srinivas et al. (2010). We regarded our theoretical result as an extension of GP-UCB algorithm analysis with the additional assumption on the smoothness of objective function and the incentive to explicitly constrain the property in the latent space. We've also added clarification of the choices of the baselines.

**[3. Problem statement]**

We provided concrete contexts to justify the necessity of applying explicit mitigation of collisions in the latent space. We pointed out the key differences between the conventional generative-model-based latent space learning and the resulted latent space of the proposed end-to-end (deep) kernel learning for Bayesian optimization.


**[4. Algorithm]**

We followed reviewer2's comments to clarified the notations used in the method section. The concrete ideas of ​​introducing the parameters and parameter choice study were added to address the concerns over the intuition and proper choice of the specific parameters in CoFLO.

Following the comments of reviwer2 and reviewer 3, we added a comparison between the regularized latent space and the non-regularized latent space in the appendix. In addition to the collisions demonstrated in the introduction section, the added figures illustrate collision's negative influence in the optimization task and **effectiveness of CoFLO** in mitigating the collision. The low-dimensional dataset we used for the illustration also demonstrates the collision's existence and impact in the dimension-reduction of **originally low-dimensional** tasks.

We merge the initial theoretical analysis section into section 4, and rephrased it as a theoretical motivation of CoFLO, to better align with the proposed contribution of this work.

**[5. Experiments]**

We have included **additional experimental results on the parameter setting study and newly introduced baseline TPE**. To address our experimental setup's earlier concern, we now include results of the statistical test and median curves.

**[6. Conclusion]**

As explained in our rebuttal, we have revised our elaboration on this work's key ideas, including additional experimental results and illustrations to highlight the proposed methods' incentive and effectiveness.

---

### Decision · Program_Chairs · 2021-01-07
**Final Decision**

**Decision:**

Reject

**Comment:**

The reviewers liked the overall idea presented in this paper. Although the idea as well as relevant tooling for incorporating constraints in the latent space has been studied a lot in the past, the authors differentiate their work by applying it in a new interesting problem. At the same time, some confusions about relation to prior work remain after rebuttal. Firstly, the theoretical additions to prior work (Srinivas et al. 2010) are still unclear in terms of significance - they feel more like observations made on top of an existing theorem rather than fresh significant insights. Furthermore, even if prior work has not considered exactly the same set-up, it would still be needed to understand what the performance would be when considering prior models or prior datasets used in similar domains (e.g. suggestions by R2, R3). The latter would be desirable especially since the experimental set-up used in this paper is deemed by the reviewers too simple (while the motivation of the paper is to solve an issue essentially manifesting in complex scenarios).